# Viscoelasticity and Solution Stability of Cyanoethylcellulose with Different Molecular Weights in Aqueous Solution

**DOI:** 10.3390/molecules26113201

**Published:** 2021-05-27

**Authors:** Qian Li, Yuehu Li, Zehua Jin, Yujie Li, Yifan Chen, Jinping Zhou

**Affiliations:** 1School of Engineering, Zhejiang A&F University, Hangzhou 311300, China; lszdnp3@163.com (Y.L.); jzh19981113@163.com (Z.J.); lyjzafu@163.com (Y.L.); yfchen.28425@foxmail.com (Y.C.); 2College of Chemistry and Molecular Science, Wuhan University, Wuhan 430072, China

**Keywords:** cyanoethylcellulose, rheological behavior, shear-thinning, solution stability

## Abstract

Water-soluble cellulose ethers are widely used as stabilizers, thickeners, and viscosity modifiers in many industries. Understanding rheological behavior of the polymers is of great significance to the effective control of their applications. In this work, a series of cyanoethylcellulose (CEC) samples with different molecular weights were prepared with cellulose and acrylonitrile in NaOH/urea aqueous solution under the homogeneous reaction. The rheological properties of water-soluble CECs as a function of concentration and molecular weight were investigated using shear viscosity and dynamic rheological measurements. Viscoelastic behaviors have been successfully described by the Carreau model, the Ostwald-de-Waele equation, and the Cox–Merz rule. The entanglement concentrations were determined to be 0.6, 0.85, and 1.5 wt% for CEC-11, CEC-7, and CEC-3, respectively. All of the solutions exhibited viscous behavior rather than a clear sol-gel transition in all tested concentrations. The heterogeneous nature of CEC in an aqueous solution was determined from the Cox–Merz rule due to the coexistence of single chain complexes and aggregates. In addition, the CEC aqueous solutions showed good thermal and time stability, and the transition with temperature was reversible.

## 1. Introduction

In recent years, natural ecosystems have been damaged due to the overuse of a wide variety of plastic polymers. The generated microplastics are difficult to completely biodegrade, triggering severe environmental issues [1]. Natural polymers as alternatives to the traditional petroleum-based materials have attracted much attention. Cellulose is the most abundant, renewable, and biodegradable natural polymer on Earth [2,3]. Cellulose-based materials, including regenerated cellulose or cellulose derivatives, have been created in the application of packaging, food, cosmetics, and environmentally friendly biocomposites. Moreover, nanocellulose obtained from natural polymers by breaking up the amorphous regions are used as reinforcement fillers for the preparation of nanocomposites. It is evident that nanocellulose has the ability to improve mechanical performance and biodegradability [4,5]. Unfortunately, due to its poor solubility in many solvents, the motivation to expand its application is still limited. To assign more properties for different needs, the chemical modification of cellulose provides an efficient approach to improve the utilization of cellulose in polymer materials [6]. Cellulose ethers (CEs), one of the most important cellulose derivatives, are produced by the reaction of cellulose with an etherification agent in the presence of aqueous NaOH. For more than 60 years, CEs have been widely used in biology, pharmaceutics, food, cosmetic, and paint industries [7,8]. However, most commercially cellulose ethers are usually followed heterogeneously by a slurry process, which yields statistically functionalized polymers. The reaction rate and degree of substitution are strongly dependent on the accessibility of the OH groups in the cellulose backbone and the interaction with the derivatizing agent [9]. The exploration of novel solvents, such as *N*,*N*-dimethylacetamide (DMAc)/lithium chloride (LiCl) [10], DMSO/tetrabutylammonium fluoride trihydrate (TBAF) [11], LiCl/1,3-dimethyl-2-imidazolidinone (DMI) [12], ionic liquids [13], and aqueous alkali system [14], offers great potential for effective homogeneous preparation of cellulose ethers. Among them, the NaOH/urea aqueous solution has been proven to be a successful, green solvent for the dissolution and etherification reaction medium of native cellulose [15]. A series of cellulose ethers such as methyl cellulose, hydroxyethyl cellulose, allyl cellulose, and methyl hydroxypropyl cellulose have been synthesized successfully in this solvent [16,17,18,19].

Notably, CE properties are sensitive to the stability and fluidity of aqueous solutions [20,21,22,23]. A detailed analysis of their rheological properties provides a basic understanding of the quality control and reliability of products. Different water-soluble polymer solutions showed various rheological properties, such as Newtonian, shear thinning, thixotropy, and gelation [24,25,26,27]. Besides the degree of substitution (DS) of CEs, the concentration, molecular weight, shear rate, temperature, and the interactions between polymer chains are the crucial parameters to understand their rheological behavior. Benchabane et al. [28] studied two critical concentrations of carboxymethylcellulose (CMC), i.e., the transition to a semidilute network solution and a concentrated solution, and found that shear thinning behavior of the CMC solution from Newtonian fluid gradually appears as concentration increases. The rheological behavior has a strong dependence on the storing time. Wang et al. [29] found that the thermodynamically reversible sol-gel transition of methylcellulose (MC) solutions has no dependence on concentration and molecular weight, but that its enthalpy change has a linear dependence on the concentration.

Cyanoethylcellulose (CEC) is an important cellulose ether which has an increased mechanical strength, heat resistance, microbiological resistance and film formation capacity [30,31]. It has been reported that CEC with low degree of substitution (DS < 1.01) is soluble in water, and that CEC samples have good solubility in organic solvents as the DS value increases to 1.37 [32]. An understanding of the molecular interaction and rheological behavior of CEC is the basis of its potential application. In previous work, the rheological properties of water-soluble CECs with high molecular weight were investigated as a function of concentration and temperature [33]. In order to provide a complete and comprehensive rheological study of water-soluble CEC in the range of low molecular weight, three water-soluble CEC samples, with different molecular weight under the same operating conditions, were synthesized through homogeneous reaction in an NaOH/urea aqueous solution by a fully homogeneous method, without the addition of a catalyst. We investigated the viscoelasticity and solution stability of CECs in the aqueous solution, and estimated the effect of concentration and molecular weight on rheological behavior. The theoretical models are used to describe the viscoelastic response for different CECs. We aim to understand more about the entanglement and association between macromolecular chains. This work provided a promising way to further extend the application of cellulose ether as a green and biodegradable thickener or emulsion stabilizer.

## 2. Results and Discussion

### 2.1. Steady-Shear Flow Behavior

Figure 1a shows the dependence of the steady shear viscosity (*η*) as a function of the shear rate for various concentrations of CEC-11 at 25 °C. The shear viscosity–shear rate curves of the CEC-11 aqueous solution exhibited a linear relationship at low concentration (c < 0.8 wt%), suggesting a feature of Newtonian fluid. The *η* has no dependence on the shear rate in the low shear rate region. The viscosity of the system increased with the increase in concentrations, due to the aggregate size and interaction between polymer chains. The shear viscosity–shear rate curves became downward at high shear rates with increasing concentration, exhibiting this as a typical shear thinning behavior, similar to the result for carboxymethyl cellulose (CMC) [34]. It also indicated that the orientation and disentanglement of the polymer were caused by the direction of the external force, resulting in a decrease in viscosity. Figure 1b shows the dependence of the shear viscosity of the 4.0 wt% CEC-3, CEC-7, and CEC-11 samples as a function of the shear rate. The shear viscosity increased as the molecular weight increased at the same concentration, suggesting the entanglement and association between macromolecular chains. Therefore, the curvature became more noticeable with an increase in concentration and molecular weight, which exhibited the increase in the pseudoplastic extent.

The Carreau model was used to fit the steady state shear behavior of the CEC solution [35]. The equation is as follows:(1)$$\eta =\frac{\eta_{0}}{{\left[1+(\lambda \gamma {)}^{n_{1}}\right]}^{(1-n_{2})/n_{1}}}$$
in which *η*_0_ is zero shear viscosity, *γ* is shear rate, *λ* is relaxation time, *n*_1_ and *n*_2_ are parameters, and *λ*, *n*_1_, and *n*_2_ do not change with *γ*. The fitting results are listed in Table 1. The Carreau model fits the CEC aqueous solution very well, and the correlation indexes are greater than 0.99. The *η*_0_ of all the CEC samples was determined successfully and showed similar trend with Figure 1. As the concentration and molecular weight increase, the viscosity increases and the relaxation time becomes shorter. The results indicate that the CEC molecular chains make contact with each other due to inter- and intramolecular hydrogen bonds, which limit the motion of molecular chains.

The zero-shear viscosity (*η*_0_) values as a function of concentration for CEC samples were plotted in Figure 2. The *η*_0_ estimated by the Carreau model increases with increasing concentration linearly at lower concentrations, which have slopes from 0.54–0.73, and turns sharply at higher concentrations. There is an obvious critical concentration point, called entanglement concentration, or *c_e_*, which represents the transition from a dilute unentangled solution to a concentrated region. When the concentration of a CEC sample is greater than *c_e_*, the *η*_0_ increases, accompanied by a very marked change. The exponents of *η*_0_ to *c* range from 3.41–3.48, due to the effective entanglements of polymers. Compared with cellulose in NaOH/thiourea aqueous solutions, the exponent of *η*_0_ on a concentration changes from 0.1–3.59 meaning the transition from a dilute solution to an entangled network [36]. Similar results are obtained for aeromonas gum in aqueous solutions [37]. Moreover, the critical concentrations of *c_e_* are 0.6, 0.85, and 1.5 wt% for CEC-11, CEC-7, and CEC-3, respectively. It could be concluded that the self-association junctions of molecular chains in the system is tighter as the length of the molecular chain increases.

It is well-known that the viscosity and the shear rate coincides a powder function at high shear velocity.
(2)$$\gamma \propto \eta^{-m}$$

The power law *−m*, and their dependency to the corresponding concentration *c* of CEC in different molecular weights, are shown in Figure 3. We can see that *−m* and *c* adopt a linear relation, and the value of *m* increases as *c* does. At a relative higher concentration, *m* is much smaller than 0.818 from the concept of Graessley [38]. This clearly indicates the occurrence of aggregation of CEC chains in aqueous solution, which is also consistent with Figure 2. As molecular weight decreases, −*m* also decreases, and the interaction among macromolecular chain is weakened.

The Ostwald-de-Waele equation is used to test the deviation extent of the CEC solutions from Newtonian fluid [39]. This equation can be written as
*σ* = *kγ**^n^*(3)
where *σ* (Pa) is shear stress, *γ* (s^−1^) is shear rate, *k* (Pa s*^n^*) is the consistency index, and *n* is the flow behavior index which reflects the degree of departure from Newtonian flow. Figure 4 shows the flow behavior index (*n*) as a function of log *k* for various concentrations at 25 °C. The whole curves can be divided into two parts. For extremely dilute CEC solutions, the values of *n* for dilute CEC solutions are approximate to 1 (*n* = 0.92–0.99), showing that the flow is close to Newtonian liquid. The solution is mainly composed of moving clusters separated from each other. At relatively higher concentrations, the *n* values decreased, and were a linear function of log *k*, suggesting that the pseudoplastic extent increased linearly with an increase in concentration and strong inter-chain interactions in the concentrated regimes. The result is similar to that of CMC in an aqueous solution; the *n* decreasing from 0.95–0.53 by 1–5% CMC concentration, revealing a strong shear thinning behavior [34]. This suggests that the network structure formed by the entanglement density between molecular chains plays an important role in the flow behavior leading to the departure from Newtonian flow. Meanwhile, the *n* values obtained by fitting the Ostwald-de-Waele equation decreased significantly with the increase in molecular weight, corresponding to the enhancement of pseudoplastic behavior. This may be due to the relatively high network of physical entanglement density caused by the polar groups in CEC-11 with a long molecular chain, which is consistent with the result in Figure 2.

### 2.2. Dynamic Oscillation Behavior

Figure 5a shows the effect of frequency on the storage modulus (*G*′) and loss modulus (*G*″) for various concentrations of CEC-11. Both *G*″ and *G*′ were strongly dependent on the frequency and the no platform area on the curve of *G’*(*w*) vs. *w*. The magnitudes of *G*″ were greater than those of *G*′ over the entire frequency range for all CEC concentrations. This means that the viscous properties are dominant compared to the elastic ones. The number of molecular chain entanglements increase with the increase in CEC concentration, which also increases the moduli. The dependence of molecular weight on dynamic oscillation behavior for 4.0 wt% of CEC-3, CEC-7, and CEC-11 is shown in Figure 5b. The *G*′ and *G*″ increase with an increase in molecular weight. Furthermore, the separation between the *G*′ and *G*″ became smaller at low frequencies as the molecular weight increased. It is difficult to form a more continuous network structure at the same concentration for low molecular weight samples with more mobile molecules.

The Cox–Merz empirical rule has been successfully applied to many polymer melting and solution systems, however it is not suitable for a polymer solution in which phase separation has occurred. Therefore, it can be used to detect the structural uniformity in a solution [40]. Figure 6 shows the Cox–Merz plot of the steady-state and dynamic viscosity of CEC-11 in an aqueous solution with different concentrations. At relatively high concentrations (*c* > 0.8 wt%), CEC-11 solutions obey the Cox–Merz law in the full measurement range (rate or frequency), i.e., complex viscosity |*η*^*^(*ω*)| = *η*(*γ*)|*_γ = ω_*. However, the data began to deviate from Cox–Merz law at lower concentrations (c < 0.8 wt%). The |*η*(ω)*| values were lower than *η (γ)* at high shear rate or frequency, which were attributed to structure decay due to the strain deformation applied to the system. The lower the solution concentration, the more obvious the deviation from the Cox–Merz rule. These results revealed structural decay as a result of coexistence of single chain inclusion complexes and aggregates in CEC solutions under the strain deformation. At high concentrations, the formation of a uniform structure due to the polymer entanglements is the main factor responsible for the whole solution. Therefore, the solution conforms to the Cox–Merz rule in the whole measurement range.

### 2.3. Solution Stability

The effects of the temperature on the rheological properties for 3 wt% CEC-11 in aqueous solution are shown in Figure 7. The curves showed viscosity behavior in the whole temperature range (from 20–80 °C), and *G*″ was larger than *G*′. The values of *G*″ and *G*′ decreased gradually with the increase in temperature. On the one hand, the kinetic energy of each moving unit increases during heating; on the other hand, with the increase in the distance between molecule chains, the free volume of the polymers increase due to thermal expansion, which is beneficial to the molecular movement. Moreover, there was no gel formation in the whole process, and tan *δ* remained almost constant. The CEC samples show good thermal stability, and the change in rheological behavior of the CEC-11 solution with temperature is reversible. It is worth noting that the curves of the heating and cooling process coincide with one another, indicating no significant thermodynamic lag in the process of heating and cooling. These entanglements are destroyed gradually at the heating process, and reformed by strong hydrogen bonding and effective interactions after cooling process.

Figure 8 shows the effects of time on the rheological properties of CEC-11 in an aqueous solution with an angular frequency of 1 rad/s. *G*″ was larger than *G*′ in the scanning time range, corresponding to the viscous fluid behavior. Additionally, the *G*″ and *G*′ increased slightly at first, and then remained unchanged as time continued. This is attributed to the initial competition of destruction and reformation of the original polymer network structure in the solution. After a certain period, the structure of the solution is in equilibrium under the dynamic oscillation. The solution maintains good stability for a long time after the dynamic time sweep, which is very important in industrial storage applications.

## 3. Materials and Method

### 3.1. Materials

Three cellulose samples (cotton linter pulp) were supplied by Hubei Chemical Fiber Group Ltd. (Xiangfan, China), and the viscosity-average molecular weight (*M*_η_) of the samples in cadoxen were determined by viscometry [41] to be 11.2 × 10^4^, 7.0 × 10^4^, and 3.1 × 10^4^, respectively. Acrylonitrile and other reagents (analytical grade) were used without further purification.

### 3.2. Homogeneous Synthesis of CEC

The CEC samples were prepared according to our previously reported method [32]. Transparent cellulose solution (4 wt%) was obtained by dissolving cellulose (cotton linter pulps) in an aqueous 7 wt% NaOH/12 wt% urea solution that precooled to −12 °C. A certain amount of acrylonitrile (the mole ratio of AGU to acrylonitrile was 1:2) was added dropwise into the 500g of cellulose solution mentioned previously, and stirred at 5 °C for 6 h. The reaction products were neutralized with acetic acid, and dialyzed with regenerated cellulose tubes (*M_w_* cutoff 8000, Union Carbide, NJ, USA) with distilled water for 7 days, and finally freeze-dried with lyophilizer (Christ Alpha 1–2, Osterode am Harz, Germany).

The purified cyanoethyl cellulose samples were labeled as CEC-11, CEC-7, and CEC-3, corresponding to the *M_η_* of cellulose. The *N*%, total DS and calculated molecular weight of CEC samples are shown in Table 2.

### 3.3. Characterization

The elemental analysis of the CECs was performed with an elemental analyzer (CHN–O-Rapid, Foss Hereaus GmbH, Hanau, Germany).

Dynamic rheology measurement was carried out on an ARES-RFS Ⅲ rheometer (TA Instruments, New Castle, DE, USA.) equipped with two force transducers, allowing for a torque measurement ranging from 0.004–1000 g·cm. Dynamic viscoelastic parameters as a function of angular frequency (*ω*), temperature (*T*), or time (*t*) were measured on a double-concentric cylinder geometry with a gap of 2 mm. The strain amplitude values were checked to ensure that all measurements were set to 10%, which is in the linear viscoelastic region. Temperature was controlled by a connection to a circulating fluid bath (Julabo FS18 cooling/heating bath) kept within ±0.2 °C of the required temperature. A thin layer of low-viscosity paraffin oil was spread on the exposed surface of the measured solution in order to prevent dehydration during rheological measurements.

The sweep of the frequency was from 0.1–100 rad/s at a constant temperature of 25 °C. The thermoreversibility of solutions were measured from 10–70 °C at *ω* = 1 rad/s and with heating or cooling rates of 1 °C/min. The steady shear viscosity was measured in the shear rate range of 0.1–1000 s^−1^. The time when the solution reached the desired temperature was defined as *t* = 0. For each measurement, a degassed CEC solution, kept at each measurement temperature without pre-shearing or oscillating, was poured into the Couette geometry instrument.

## 4. Conclusions

The rheological properties of water-soluble CEC samples in different molecular weights and concentrations were studied by rheometer. The transition from Newtonian fluid to shear thinning behavior appeared with an increase in the concentration, and *η* has a strong dependence on concentration due to the aggregate size and interaction between polymer chains. With increasing molecular weight, the pseudoplastic extent became obvious at a lower shear rate under the same conditions. The magnitudes of *G*″ are superior to *G*′ for all concentrations indicating liquid-like behavior. The deviation from the Cox–Merz law for CEC solutions is related to heterogeneous structures in dilute solutions, and the flow behavior obeys the Cox–Merz law with the increase in CEC concentration. In addition, CEC solutions showed good thermal stability and temperature reversible viscosity behavior, with no significant effect on the solution properties after long-term storage. We hope our work will contribute meaningful information to flow behavior characteristics which are required for the design and operation of these water-soluble cellulose ethers in industry-scale application.

## Figures and Tables

**Figure 1 molecules-26-03201-f001:**
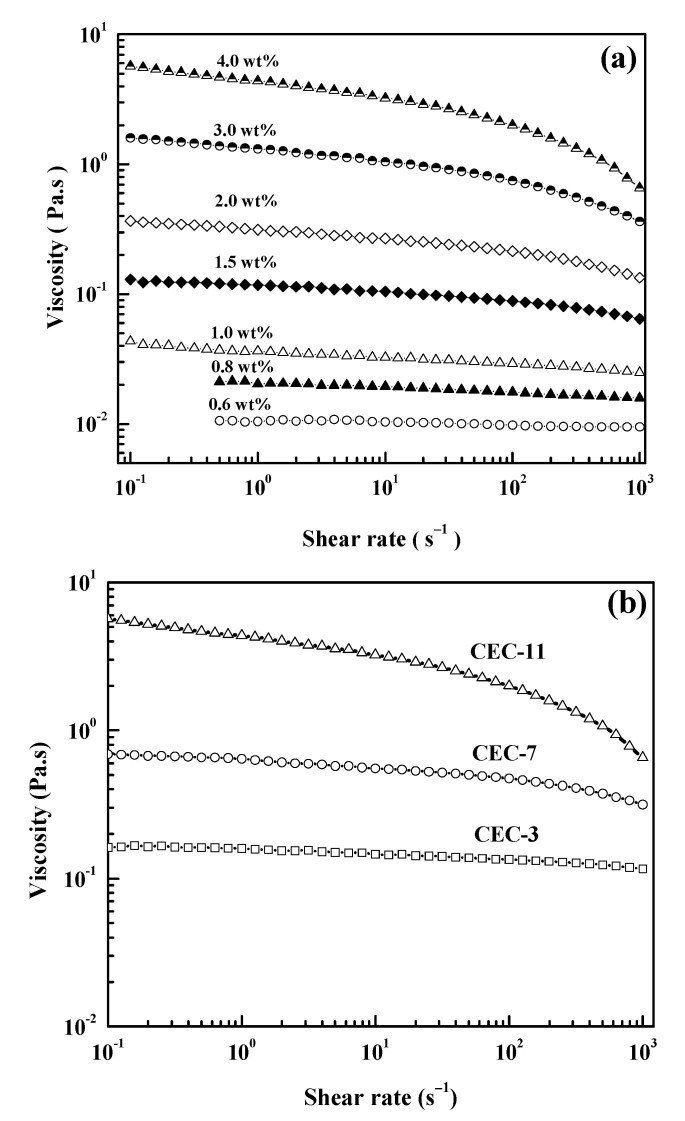
Dependence of the steady viscosity on the shear rate at 25 °C: (**a**) for various concentrations of CEC-11, (**b**) for 4.0 wt% of CEC-3, CEC-7, and CEC-11.

**Figure 2 molecules-26-03201-f002:**
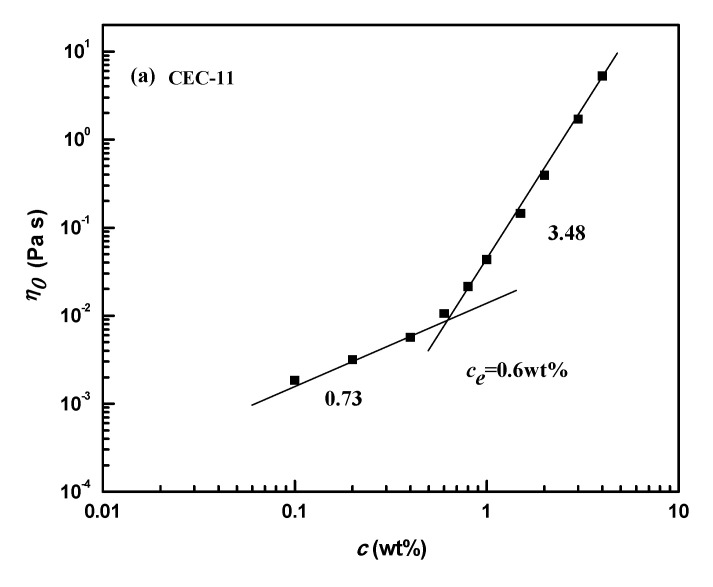
The zero-shear viscosity (*η*_0_) values as a function of concentration for CEC samples.

**Figure 3 molecules-26-03201-f003:**
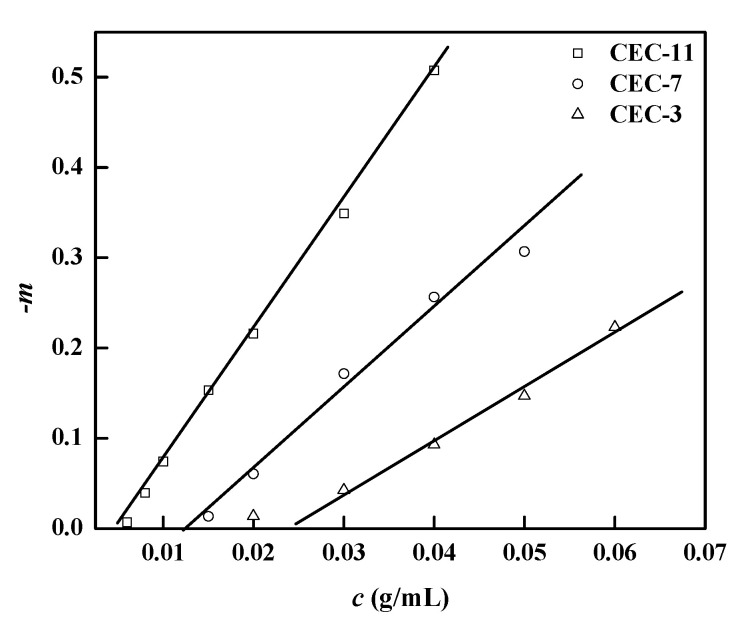
Slopes in the pseudoplastic domain of *η*(*γ*) determined on flow curves for various concentrations of CEC-11, CEC-7 and CEC-3 solutions.

**Figure 4 molecules-26-03201-f004:**
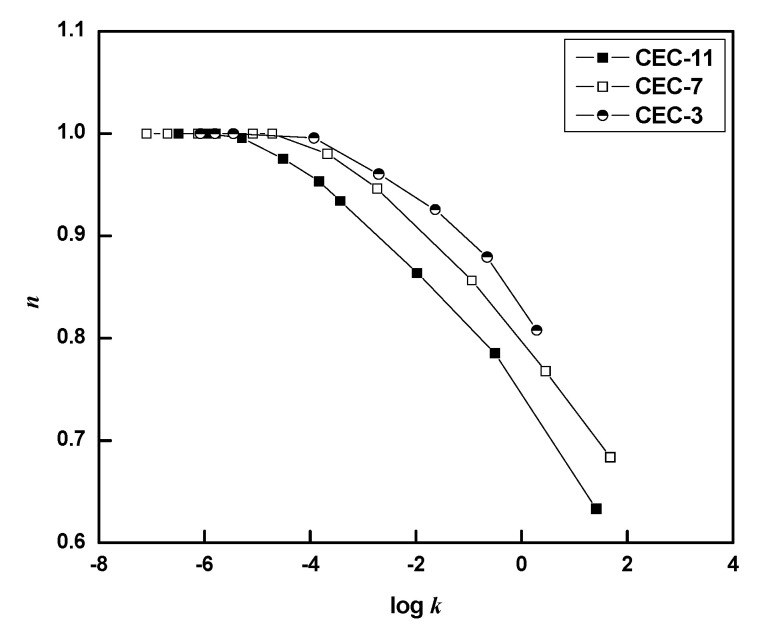
Flow behavior index (*n*) as a function of log *k* for various concentrations of CEC samples.

**Figure 5 molecules-26-03201-f005:**
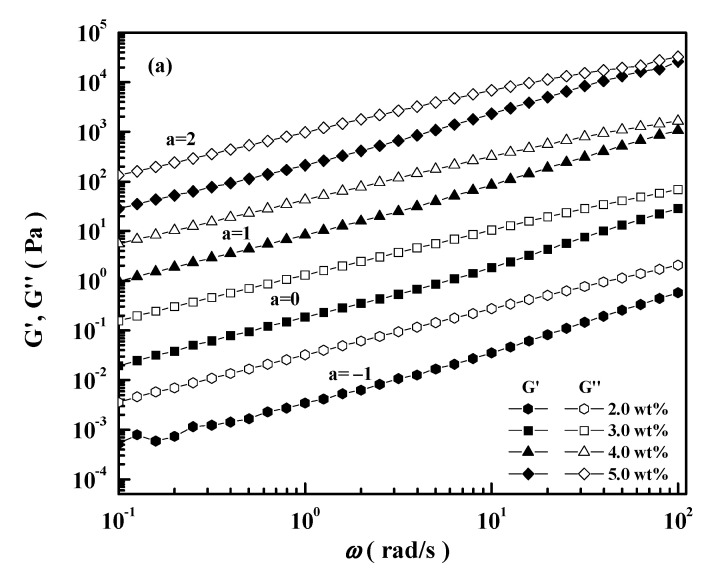
*G*′ and *G*″ as a function of angular frequency at 25 °C: (**a**) for various concentrations of CEC-11; (**b**) for 4.0 wt% of CEC-3, CEC-7, and CEC-11.

**Figure 6 molecules-26-03201-f006:**
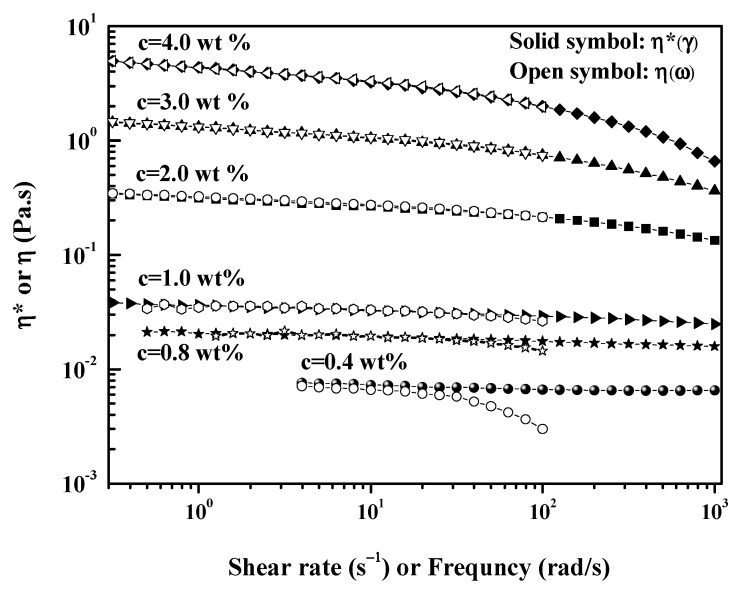
Cox–Merz plots of CEC-11 in distilled water at 25 °C.

**Figure 7 molecules-26-03201-f007:**
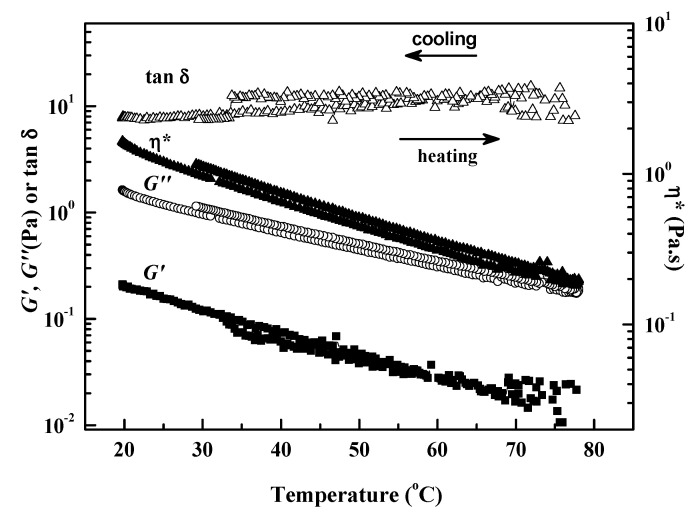
Influences of the temperature on the rheological properties of CEC-11 in aqueous solution.

**Figure 8 molecules-26-03201-f008:**
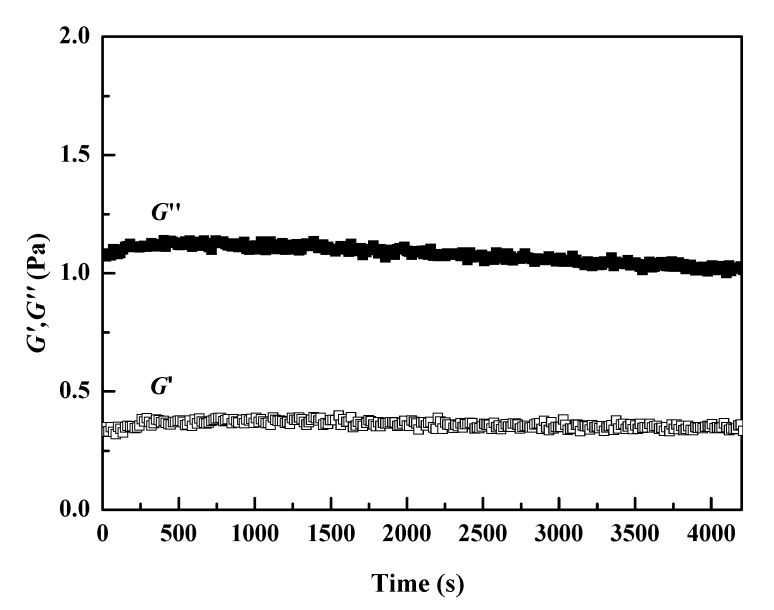
Influence of time on the rheological properties of CEC-11 in aqueous solution.

**Table 1 molecules-26-03201-t001:** Fitting results of Carreau model for CEC samples.

Sample	*c* (wt%)	*η*_0_ (Pa s)	*λ* (s)	*n* _1_	*n* _2_	*R* ^2^
CEC-11	5.0	8.346	1.936 × 10^−5^	0.28	0.68	0.992
	4.0	5.824	8.644 × 10^−4^	0.35	0.57	0.991
	3.0	1.689	6.791 × 10^−3^	0.33	0.54	0.993
CEC-7	5.0	1.753	0.016	0.41	0.71	0.997
	4.0	0.722	9.581 × 10^−3^	0.32	0.64	0.991
	3.0	0.265	5.70 × 10^−3^	0.27	0.83	0.997
CEC-3	5.0	0.274	0.146	0.18	0.57	0.994
	4.0	0.186	3.48 × 10^−3^	0.28	0.63	0.995
	3.0	0.123	0.0128	0.15	0.91	0.992

**Table 2 molecules-26-03201-t002:** The *N*%, total DS and calculated molecular weight of CEC samples.

Samples	*N*%	Total DS ^*a*^	Molecular Weight *^b^*
CEC-11	2.98	0.39	12.63 × 10^4^
CEC-7	3.12	0.41	7.94 × 10^4^
CEC-3	3.36	0.44	3.54 × 10^4^

*^a^* Calculation of DS based on nitrogen content of the samples according to our previous work [32]: DS = (162 × *N*%)/(1400 − 53 × *N*%). *^b^* When estimating molecular weight of CEC, we assumed no degradation of macromolecular chain occurred during chemical reaction. Each substitution of OH by O-CH_2_-CH_2_-CN group increased the molecular weight of one anhydrous glucose residue from 162 to 215 g/mol. Therefore at certain DS, theoretically, the molecular weight varied to *M* × ((162 + 53 ∗ DS)/162).

## Data Availability

The data presented in this study are available on request from the corresponding author.

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
