# Peer review of "Viscoelasticity and Solution Stability of Cyanoethylcellulose with Different Molecular Weights in Aqueous Solution"

_molecules, 2021, doi:10.3390/molecules26113201_

Round 1

Reviewer 1 Report

The manuscript deals with the rheology of aqueous solutions of cyanoethylcellulose. The results correspond to those of other water-soluble cellulose derivatives. Even being of limited novelty, such reports might find readers if carried out and presented properly.  Unfortunately, this is not the case of this manuscript. Authors give no comparison with literature date on rheology of other water-soluble cellulose derivatives; on the other, they duplicitously present the same data – Fig. 1 and Fig. 3 give practically the same information. It is a question whether a sufficiently wide range of concentrations was explored. Nevertheless, the omission which makes the data worthless is the absence of proper molecular characterization of the cyanoethylcellulose samples. Molecular weight is assigned to that of the parental cellulose only, ignoring the effect of the substitution and subsequent dialysis. Nothing is known about the degree of substitution.

Author Response

Reviewer-1

The manuscript deals with the rheology of aqueous solutions of cyanoethylcellulose. The results correspond to those of other water-soluble cellulose derivatives. Even being of limited novelty, such reports might find readers if carried out and presented properly. Unfortunately, this is not the case of this manuscript.

Authors give no comparison with literature date on rheology of other water-soluble cellulose derivatives; on the other, they duplicitously present the same data – Fig. 1 and Fig. 3 give practically the same information. It is a question whether a sufficiently wide range of concentrations was explored. Nevertheless, the omission which makes the data worthless is the absence of proper molecular characterization of the cyanoethylcellulose samples. Molecular weight is assigned to that of the parental cellulose only, ignoring the effect of the substitution and subsequent dialysis. Nothing is known about the degree of substitution.

Response:

  1. We thank the reviewer for the very relevant comment. We have compared our resultswith other water-soluble carboxymethyl cellulose (CMC), cellulose and aeromonas gum in aqueous solutions in the reference [34], [36] and [37], the corresponding text of which has been revised to the following.

For Figure 1, “The shear viscosity-shear rate curves became downward at high shear rate with increasing concentration, exhibiting this is a typical shear-thinning behavior, similar to the result for carboxymethyl cellulose (CMC) [34]”.

For Figure 2, “Compared with cellulose in NaOH/thiourea aqueous solutions, the exponent ofη0 on concentration changed from 0.1 to 3.59 means the transition from a dilute solution to an entangled network [36]. The similar results are obtained for aeromonas gum in aqueous solutions [37]”

For Figure 4, “The result is similar with CMC in aqueous solution, the n decrease from 0.95 to 0.53 for 1-5 % CMC concentrations, revealing a strong shear thinning behavior [34].”

  1. Moreover, we have changed Figure 3 to new Figure 3.
  2. We have investigated the rheological properties as a function of concentration in the range of 0.1-6 wt% and find all the results follows the theoretical rules.
  3. Dialysis process were carried out using the dialysis tubes with Mwcutoff 8000. This means the dialysis process would only allow the remove of molecules with Mw less than 8000. Considering that CEC in our study all have the Mw above 3 x 104, we can confirmed that the dialysis would have negligible influence on molecular weight of CEC.

       By following the reviewer’s advice, we have added the N %, total DS, and calculated molecular weight of CEC samples as Table 2. 

Table 2 The N %, total DS and calculated molecular weight of CEC samples.

Samples

N %

Total DSa

molecular weight b

CEC-11

2.98

0.39

12.63×104

CEC-7

3.12

0.41

7.94×104

CEC-3

3.36

0.44

3.54×104

a, Calculation of DS based on nitrogen content of the samples according to our previous work [32]: DS = (162 × N %)/(1400 – 53 × N %)

b, When estimated of molecular weight of CEC samples, we assumed no degradation of macromolecular chain occurred during chemical reaction. Each substitution of OH by O-CH2-CH2-CN group increased the molecular weight of one anhydrous glucose residue from 162 to 215 g/mol. Therefore at certain DS, theoretically, the molecular weight varied to M × ((162 + 53*DS) / 162).

Reviewer 2 Report

Dear Authors,

Generally, this research is very interesting because it broadens the knowledge about cellulose and its processing. I also appreciate the fact that the authors probably put a lot of work into the preparation of this manuscript. I also believe that the strengths of this article are the research results. However, the article has some imperfections that need to be corrected. My comments mainly concern the introduction, discussion of the results and conclusions.

Detailed comments below:

  1. Introduction: I think that at the beginning of the introduction it is worth adding information about new pro-ecological applications of cellulose. Currently, many biodegradable products, e.g. biocomposites, are created on the basis of cellulose. There is also a lot of research carried out, where cellulase is used as a component that strengthens the strength of biocomposites. You can also mention other interesting applications. Such a written introduction will greatly emphasize the importance of your research. Read the following articles on this topic: “Properties of Biocomposites from Rapeseed Meal, Fruit Pomace and Microcrystalline Cellulose Made by Press Pressing: Mechanical and Physicochemical Characteristics”; “The rape pomace and microcrystalline cellulose composites made by press processing”; “ Valorisation of vine shoots for the development of cellulose-based biocomposite films with improved performance and bioactivity”. Continue reading the introduction. I believe that an additional description should be added before formulating the purpose of the research. I am thinking of better emphasizing the importance of your research (novelty, innovation). Perhaps it is worth pointing to future applications of your research (or products).
  2. Results and Discussion: I think that the description of the research results is well done, but the article should better conduct the research discussion. I mean mainly applying more references to the research of other authors. In the discussion of the results, it is also worth comparing the obtained results with commercial applications (if, of course, there are any). Then it is easier to compare and interpret the test results. Try to add some new literature.

Table 1. For the coefficient of determination R2, 3 decimal places are enough.

In Figure 3, Figure 2, the graphs show no signs: a, b, c, d. Is this a mistake? You will also notice that the figure 2 lacks a unit on the Y axis. This should complete each other.

Experimental Section: I admit that I prefer the name "Materials and Method". This, in my opinion, is also a requirement of the journal. This part of the article should precede the research results. Therefore, I am asking you to place the research methodology in accordance with the guidelines of the journal.

(description of research equipment). I think it will be a little better: model (manufactor: …… .., country, city)

Conclusion: I think one more conclusion should be added. This will make it easier to answer the question. What do your research contribute to, for example, improve the production of new materials, improve a technology, etc. Overall, this part of the conclusions may be more general (it may also be forward-looking).

Author Response

Dear Authors,

Generally, this research is very interesting because it broadens the knowledge about cellulose and its processing. I also appreciate the fact that the authors probably put a lot of work into the preparation of this manuscript. I also believe that the strengths of this article are the research results. However, the article has some imperfections that need to be corrected. My comments mainly concern the introduction, discussion of the results and conclusions.

Detailed comments below:

  1. Introduction:I think that at the beginning of the introduction it is worth adding information about new pro-ecological applications of cellulose. Currently, many biodegradable products, e.g. biocomposites, are created on the basis of cellulose. There is also a lot of research carried out, where cellulase is used as a component that strengthens the strength of biocomposites. You can also mention other interesting applications. Such a written introduction will greatly emphasize the importance of your research. Read the following articles on this topic: “Properties of Biocomposites from Rapeseed Meal, Fruit Pomace and Microcrystalline Cellulose Made by Press Pressing: Mechanical and Physicochemical Characteristics”; “The rape pomace and microcrystalline cellulose composites made by press processing”; “ Valorisation of vine shoots for the development of cellulose-based biocomposite films with improved performance and bioactivity”. Continue reading the introduction. I believe that an additional description should be added before formulating the purpose of the research. I am thinking of better emphasizing the importance of your research (novelty, innovation). Perhaps it is worth pointing to future applications of your research (or products).

Response: We appreciate the reviewer 2 for his/her comments and suggestion as well as for sharing his/her view that our work broadens the knowledge about cellulose and its processing. We learned a lot after careful read of the recommended articles. A detailed, new description about new pro-ecological applications of cellulose and the relevant references (new reference [1], [3 4]) have been added in the Introduction. Moreover, we highlighted the importance of our research and rewrote the potential future applications of CEC based on our research. All the corresponding revisions are marked blue in our manuscript.

  1. Results and Discussion:I think that the description of the research results is well done, but the article should better conduct the research discussion. I mean mainly applying more references to the research of other authors. In the discussion of the results, it is also worth comparing the obtained results with commercial applications (if, of course, there are any). Then it is easier to compare and interpret the test results. Try to add some new literature.

Response: Thanks for such kind suggestion. We have add some new references of other authors and compare the obtained results about carboxymethyl cellulose (CMC), cellulose and aeromonas gum in aqueous solutions for Figure 1, 2 and 4. The corresponding new references [34], [36], [37] are added. The revised are as follows: 

For Figure 1, “The shear viscosity-shear rate curves became downward at high shear rate with increasing concentration, exhibiting this is a typical shear-thinning behavior, similar to the result for carboxymethyl cellulose (CMC) [34]”.

For Figure 2, “Compared with cellulose in NaOH/thiourea aqueous solutions, the exponent of η0 on concentration changed to 3.59 means the transition from a dilute solution to an entangled network [36]. The similar results are obtained for aeromonas gum in aqueous solutions [37]”

For Figure 4, “The result is similar with CMC in aqueous solution, the n decrease from 0.95 to 0.53 for 1-5 % CMC concentrations, revealing a strong shear thinning behavior [34].”

  1. Table 1. For the coefficient of determination R2, 3 decimal places are enough.

Response: Following the reviewer’s advice, the decimal places of R2 have been changed to 3.

  1. In Figure 3, Figure 2, the graphs show no signs: a, b, c, d. Is this a mistake? You will also notice that the figure 2 lacks a unit on the Y axis. This should complete each other.

Response: Thanks for the good advice. This is indeed a mistake. We have added signs and unit in Figure 2. According to another reviewer’s suggestion, we have changed Figure 3 to new Figure 3.

  1. Experimental Section:I admit that I prefer the name "Materials and Method". This, in my opinion, is also a requirement of the journal. This part of the article should precede the research results. Therefore, I am asking you to place the research methodology in accordance with the guidelines of the journal.

Response: Thanks for the reviewer’s kind suggestion. We change the “Experimental Section” to “Materials and Method”. According to the template of Molecules, the part 2, part 3 and part 4 are “Results”, “Discussion” and “Materials and Methods”, respectively. Thus the structure of the article is preserved.

  1. (description of research equipment). I think it will be a little better: model (manufactor: …… .., country, city)

Response: We have added the detailed description of rheometer including city, state and so on.

  1. Conclusion:I think one more conclusion should be added. This will make it easier to answer the question. What do your research contribute to, for example, improve the production of new materials, improve a technology, etc. Overall, this part of the conclusions may be more general (it may also be forward-looking).

Response: Thanks for the good advice. We have added the contribution of our research in “Conclusion” section according to the reviewer’s suggestion and now reads “We hope our work will contribute meaningful information to flow behavior characteristics which are required for the design and operation of these water soluble cellulose ethers in industry-scale application”.

Round 2

Reviewer 1 Report

The authors added the values of degree of substitution and thus removed my main objection to the publication of the manuscript. Nevertheless, some minor correction still have to be made:

Table 2 - 104 NOT 104

Newly added text on page (after [33.] is a nonsensical fragment, not a sentence.

A power function NOT powder (below Figure 2).

interactions between macromolecular chains NOT interaction among macromolecular chain (page 4 last line)

Author Response

The authors added the values of degree of substitution and thus removed my main objection to the publication of the manuscript. Nevertheless, some minor correction still have to be made:

1. Table 2 - 104 NOT 104

2. Newly added text on page (after [33.] is a nonsensical fragment, not a sentence.

3. A power function NOT powder (below Figure 2).

4. interactions between macromolecular chains NOT interaction among macromolecular chain (page 4 last line)

Response:

Thank you very much for the reviewer’s good evaluation and kind advice. Your suggestion has given us a lot of help. The last version has been revised as follows:

  1. In Table 2, the “104” has been changed to “104
  2. Thanks for the good suggestions! The sentence has been corrected. The new sentence is “In order to provide a complete and comprehensive rheological study of water-soluble CEC in the range of low molecular weight, three water-soluble CEC samples with different molecular weight were synthesized through homogeneous reaction in NaOH/urea aqueous solution by a fully homogeneous method without extra addition of catalyst.”
  3. “powder ” has been corrected to “power”
  4. “interaction among macromolecular chain” has been corrected to “interactions between macromolecular chains”

I expressed my sincere thanks for your kind help and good suggestions.

Reviewer 2 Report

Dear Authors,

I accept the corrections made in this manuscript. In my opinion, the manuscript can be published as is. 

A little note. The methodology section should be placed before the description of the research results.

Author Response

Dear Authors,

I accept the corrections made in this manuscript. In my opinion, the manuscript can be published as is. A little note. The methodology section should be placed before the description of the research results.

Response:

Dear Reviewers,

Thanks for your kindness and patience in dealing with our submission. We have initially followed the traditional format of the publication template of Molecules journal, thus taken the <Materials and Method> section just behind the <Results and Discussion> section. Since both the style you suggested and the present one match the journal, we decide not to change the structure. Thanks for your suggestions again.